# Laser–Plasma Wake Velocity Control by Multi-Mode Beatwave Excitation in a Channel

Alexander Pukhov [1,*], Nikolay E. Andreev [2,3], Anton A. Golovanov [4,5], Ivan I. Artemenko [5] and Igor Yu Kostyukov [5]

1 Institut für Theoretische Physik I, Universität Düsseldorf, 40225 Düsseldorf, Germany
2 Joint Institute for High Temperatures of the Russian Academy of Sciences, 125412 Moscow, Russia
3 Moscow Institute of Physics and Technology, State University, 141701 Moscow, Russia
4 Department of Physics of Complex Systems, Weizmann Institute of Science, Rehovot 7610001, Israel
5 Department of Radiophysics, Lobachevsky State University of Nizhny Novgorod, 603950 Nizhny Novgorod, Russia
* Correspondence: pukhov@tp1.hhu.de

**Abstract:** The phase velocity of a laser-driven wakefield can be efficiently controlled in a plasma channel. A beatwave of two long laser pulses is used. The frequency difference between these two laser pulses equals the local plasma frequency, so that the slow resonant excitation of the plasma wave is possible. Because the driver energy is spread over many plasma periods, the interference pattern can run with an arbitrary velocity along the channel and generate the wakefield with the same phase velocity. This velocity is defined by the channel radius and the structure of laser transverse modes excited in the channel. The wake velocity can be matched exactly to the witness velocity. This can be the vacuum speed of light for ultra-relativistic witnesses, or subluminal velocities for low-energy, weakly relativistic witnesses such as muons.

**Keywords:** laser–plasma acceleration; optical modes; plasma channel; wakefield velocity control; muon acceleration





## 1. Introduction

Proposed some 40 years ago, plasma wakefield acceleration [1] can produce gradients hundreds of times higher than those achievable with conventional techniques based on radiofrequency cavities [2,3]. In this concept, a strong driver—a high-charge particle bunch or a relativistically intense laser pulse—excites a plasma wave (wakefield). The plasma wave has a longitudinal component of the electric field, and thus is suitable for particle acceleration. An externally injected or even self-trapped witness bunch can remain in the accelerating phase of the wakefield over a long propagation distance and gain energy. Plasma wakefield experiments have demonstrated impressive energy gains over distances much shorter than those in the conventional accelerators. In the SLAC afterburner experiment [4], the team directed a 42 GeV beam from the linac into lithium gas in an 85 cm long plasma chamber. Measurements showed that some electrons more than doubled their energy, up to a maximum of 85 ± 7 GeV, implying a peak accelerating field of around 53 GV/m. In laser wakefield experiments, multi-GeV energies have been reported [5–7], with peak accelerating fields around 100 GV/m [8,9].

The range of interest for high-energy particle physics extends to energies well beyond 10 TeV. As conventional accelerators based on solid-state radiofrequency resonator technology can provide fields around 100 MV/m, future collider machines built on this technology would have monstrous dimensions of tens of kilometers. The use of plasma can make these accelerators much more compact. However, plasma-based accelerators also have limitations. Laser wakefield acceleration in plasmas is limited by several physical effects: (i) laser pulse energy depletion, (ii) laser pulse diffraction, (iii) dephasing, or (iv) radiation losses.

To overcome the pulse diffraction, so that the laser can propagate many Rayleigh lengths, one can either employ relativistic self-guiding in the bubble regime [10] or use a pre-formed guiding plasma channel [11–15].

Another crucial problem is the dephasing of relativistic particles in laser-driven wakefields. Typical $\gamma$-factors associated with the laser pulse group velocity in plasma are in the range $\gamma_w = 10 \ldots 30$. The wakefield phase velocity usually is close to the laser group velocity. The dephasing length of a relativistic ($\gamma \gg \gamma_w$) witness then scales as $L_d \propto \lambda_p \gamma_w^2$, where $\lambda_p$ is the plasma wavelength.

Plasma waves not only have strong accelerating longitudinal fields, but also focusing transverse fields. Light particles such as electrons and positrons experience transverse betatron oscillations in these fields. At very high energies, they radiate, and this radiation may limit the maximum energy achievable in the plasma wakefield [16,17].

One of the ways to overcome the problem of radiative losses is to accelerate heavier particles: the radiated power scales as $(m_e/M)^4$, where $m_e$ is the electron mass and $M$ is the mass of particles to be accelerated. One of such candidates are muons, which are point-like particles [18] and leptons such as electrons and positrons, but have the mass $Mc^2 = 105.66$ MeV, which corresponds to the mass ratio $M/m_e \approx 206$. However, they are unstable particles with a lifetime of $\tau_\mu \approx 2.2\,\mu$s. Because of this, muons have to be accelerated to highly relativistic energies in a much shorter time, which might mandate the use of high-gradient plasma wakefield accelerators.

Muons can only be obtained through the decay of pions produced in nuclear interactions between accelerated particles and nuclear targets. Presently, high-intensity proton accelerators are the most popular sources of accelerated particles used to produce muons. The produced muons initially are only weakly relativistic ($\gamma \approx 2 \ldots 3$), so it is difficult to trap and accelerate them in usual plasma wakes generated by a highly relativistic driver such as a laser pulse or charged particle bunch: slow muons fall back out of the acceleration phase of the wake rather fast. High-amplitude wakefields also cannot self-trap muons because of their high mass.

In this work, we set a goal of controlling the plasma wakefield phase velocity. To achieve it, we propose to use beatwave wakefield excitation via a mixture of transverse modes in a plasma channel. The original idea of laser wakefield acceleration proposed by Tajima and Dawson [1] considered resonant beatwave plasma wake excitation by a single pulse. The resulting wake still has a phase velocity close to the laser group velocity in uniform plasma. Here, we show that the interference pattern of two different transverse laser modes with different frequencies in a plasma channel can produce a wake with a phase velocity controllable in a broad range of values. It can be tuned to be exactly equal to the speed of light, become superluminal, or match the velocity of a weakly relativistic muon bunch. This allows the acceleration particles resonantly with no dephasing.

We provide here the linear theory of this regime as well as particle-in-cell simulations. We stress that nonlinear effects may limit the acceleration process at later times. These will be considered in a separate study.

## 2. Materials and Methods

### 2.1. Multi-Modal Wakefield Excitation

Let us consider two electromagnetic waves in a parabolic plasma channel or capillary. These two waves have different frequencies, $\omega_1$ and $\omega_2$. For the resonant excitation of a plasma wave, these two waves have to satisfy the condition $\omega_1 = \omega_2 + \omega_p$. Let these two waves be different transverse modes with the transverse wavenumbers $k_{1\perp}$ and $k_{2\perp}$. In the linear approximation, the dispersion relationship gives

$$\omega_i^2 = c^2 k_{i\parallel}^2 + c^2 k_{i\perp}^2 + \omega_p^2 = c^2 k_{i\parallel}^2 + (\Delta\omega_i)^2, \ \ i = 1, 2, \tag{1}$$

where we introduce the notation $\Delta\omega_i = \sqrt{c^2 k_{i\perp}^2 + \omega_p^2}$.

Equation (1) is rather general, and is valid for plasma channels of arbitrary geometry. We can make it more concrete by selecting the particular transverse plasma profile.

(i) For the case of a cylindrical plasma capillary of radius $R$ filled with a uniform plasma, the transverse mode structure is defined by zero-order Bessel functions $J_0(k_{i\perp n}r)$, where the transverse wave mode number $k_{i\perp n}$ with $n = 1, 2, 3, \ldots$ is

$$k_{i\perp n} = \frac{u_n}{R}\left(1 - i\frac{\mu}{k_i R}\right). \tag{2}$$

Here, $k_i = \omega_i/c$ is the vacuum wavenumber, and $u_n$ is the $n^{\text{th}}$ root of the Bessel function: $J_0(u_n) = 0$. The parameter $\mu = (\varepsilon_w + 1)/2\sqrt{\varepsilon_w - 1}$ defines leakage/absorption through the capillary boundaries with the dielectric constant $\varepsilon_w$ [19].

For simplicity, in the following discussion we neglect the imaginary parts responsible for the laser absorption or leakage through the channel walls.

(ii) In a plasma channel with parabolic radial density distribution

$$n_e(r) = n_0\left[1 + \left(\frac{r}{R_{ch}}\right)^2\right] \tag{3}$$

the transverse field structure is defined by the basic Laguerre–Gauss functions

$$D_n(\rho) = L_n(\rho)\exp(-\rho/2), \quad \rho = k_p r^2/R_{ch}, \tag{4}$$

where $k_p = \omega_{p0}/c = \omega_p(r = 0)/c$. The corresponding transverse wavenumber for the mode $n$ $(n = 1, 2, 3, \ldots)$ is

$$k_{i\perp n} = \frac{\omega_{p0}}{\omega_i}\left(\frac{2n+1}{k_p R_{ch}}\right)^{1/2}. \tag{5}$$

(iii) The simplest analytics we obtain for a two-dimensional plasma waveguide in a slab geometry; in this case, the transverse field structure is given by harmonic functions $\cos((2n-1)k_\perp r)$ for symmetric modes and $\sin(2nk_\perp)$ for antisymmetric modes $(n = 1, 2, 3, \ldots)$. Here,

$$k_\perp = \frac{\pi}{d} \tag{6}$$

with $d$ being the distance between the waveguide walls.

### 2.2. Wakefield Phase Velocity

The wakefield phase velocity $v_w$ is determined by the velocity of the low-frequency envelope of the two waves:

$$\frac{v_w}{c} = \frac{\omega_1 - \omega_2}{c\left(k_{1\parallel} - k_{2\parallel}\right)}. \tag{7}$$

We assume that the plasma inside is sufficiently underdense ($\omega_p \ll \omega_i$) and the channel is wide ($k_i d \gg 1$). In this case, $(\Delta\omega_i)^2 \ll \omega_i(\omega_1 - \omega_2)$, and we can develop

$$\frac{v_w}{c} \approx 1 + \frac{1}{2(\omega_1 - \omega_2)}\left[\frac{(\Delta\omega_1)^2}{\omega_1} - \frac{(\Delta\omega_2)^2}{\omega_2}\right]. \tag{8}$$

Equation (8) gives us the option to control the wake phase velocity by choosing the channel parameters and the mode structure.

First of all, we can exactly match it to the speed of light, $v_w = c$, if we select

$$\frac{(\Delta\omega_1)^2}{\omega_1} = \frac{(\Delta\omega_2)^2}{\omega_2}. \tag{9}$$

This translates into

$$k_{1\perp}^2 = \frac{\omega_1}{\omega_2}k_{2\perp}^2 + \frac{\omega_1 - \omega_2}{\omega_2}\frac{\omega_p^2}{c^2}. \tag{10}$$

In general, if we want a certain deviation of the wake phase velocity from the speed of light, $\delta\beta = v_w/c - 1$, and resonantly excite the plasma wakefield, so that $\omega_1 = \omega_2 + \omega_p$, the transverse wavenumbers of the modes must satisfy the relationship

$$c^2 k_{1\perp n}^2 = \left(c^2 k_{2\perp m}^2 + \frac{\omega_p^3}{\omega_1}\right)\left(1 + \frac{\omega_p}{\omega_1}\right) + 2\omega_1\omega_p\delta\beta. \tag{11}$$

Choosing proper mode numbers and matched parameters of the waveguiding structure, one can control the wakefield phase velocity in a broad range and accelerate the wake or slow it down according to the actual witness velocity.

## 3. Results

To demonstrate the validity of the analytical theory, we perform a two-dimensional (2D) particle-in-cell (PIC) simulation in a slab geometry using the code VLPL [20]. The 2D slab geometry has been chosen to keep the simulation costs at a reasonable level. In the simulation, we assume two linear laser pulses with wavelengths $\lambda_1 = 770$ nm and $\lambda_2 = 855$ nm. Both pulses have a Gaussian longitudinal profile with a FWHM duration of $\tau = 140$ fs and peak intensity of $I_1 = I_2 = 6.8 \times 10^{16}$ W/cm$^2$, corresponding to the dimensionless relativistic laser amplitudes $a_1 = 0.2$ and $a_2 = 0.22$. The simulation box is 10 μm × 200 μm in size and contains a numerical grid of 20 cells per laser wavelength $\lambda_1$.

The interference patterns of the electromagnetic field inside the waveguide at different propagation distances are shown in Figure 1. As the laser pulses propagate through the plasma, the peaks of the interference pattern shift back along the laser pulses of the co-moving coordinate, which corresponds to their subluminal velocity. These laser pulses excite the longitudinal wakefield shown in Figure 2, which also demonstrates the subluminal phase velocity of the excited wake.

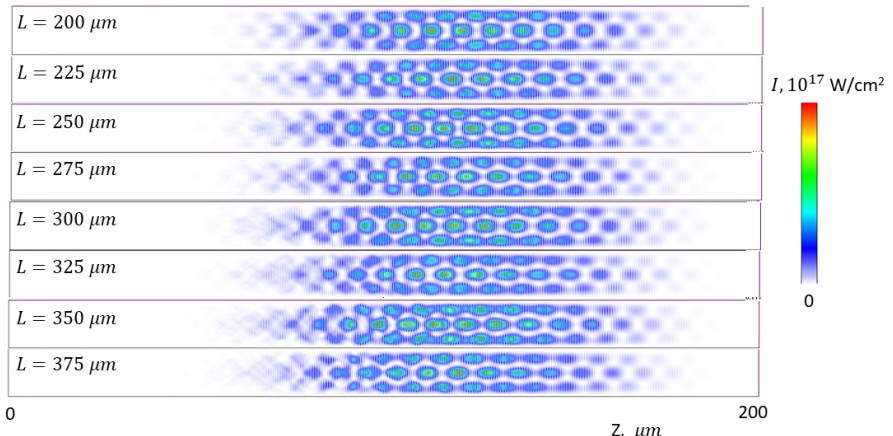

**Figure 1.** Interference pattern of the laser intensity in the co-moving window at different propagation distances. The transverse size of the simulation box is 10 μm.

We take a plasma waveguide with the transverse size $d = 8.4$ μm filled with plasma of uniform electron density $n_e = 1.86 \times 10^{19}$ cm$^{-3}$. The channel wall density is $n_{wall} = 10^{21}$ cm$^{-3}$. We used eight numerical macroparticles per cell to simulate the plasma. Laser pulse #1 is excited at the ground symmetric mode $n = 1$, while laser pulse #2 is excited at the next symmetric transverse mode with $n = 2$. These parameters correspond to an excited plasma wakefield with $v_w = 0.9c$ in the linear limit. The chosen numerical parameters are rather artificial, and are to demonstrate the principle of wakefield velocity control while keeping the computational costs at a reasonable level.

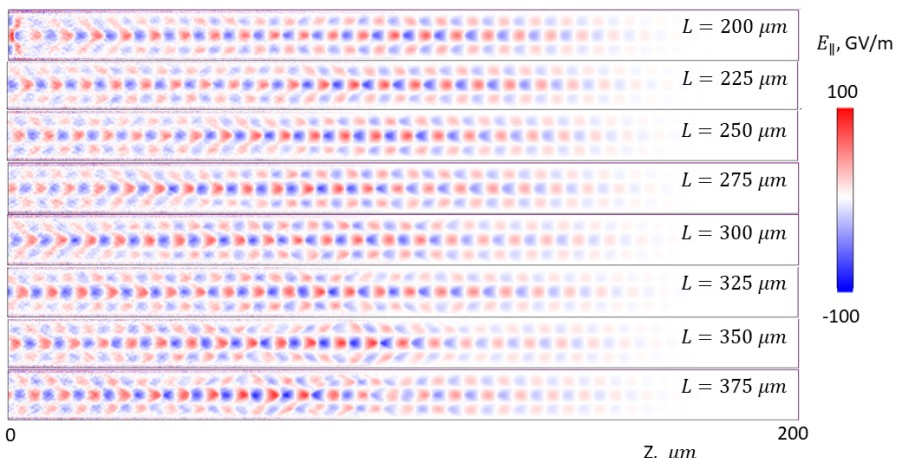

**Figure 2.** Longitudinal accelerating field at different propagation distances. The wakefield is resonantly excited by the beating of the two laser pulses.

To verify the possibility of acceleration without dephasing, we inject a test muon bunch with the initial $p_\parallel = 2.2Mc$ (corresponding to $v_\parallel = 0.9c$, equal to the expected wake phase velocity) and the sizes $\sigma_\parallel = 120$ μm and $\sigma_\perp = 20$ μm. The muon density is considered to be very low in these simulations, so we have no beam loading of the wakefield. The evolution of the witness bunch density is shown in Figure 3. The longitudinal phase space $\left(p_\parallel, z\right)$ is given in Figure 4. We can see that the relatively slow muons are resonantly accelerated by the subluminal wakefield. As they gain significant longitudinal momentum, they start to overtake the wake, and the rate of their energy gain reduces. To overcome this problem, one could use a tapered plasma waveguide. However, this topic is out of the scope of the present work, and will be studied in further publications.

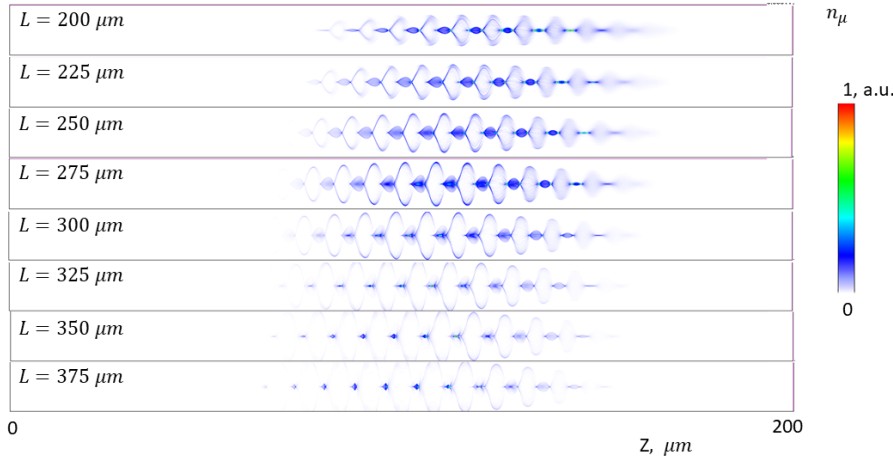

**Figure 3.** Evolution of the witness bunch (muon) density showing resonant focusing/defocusing of the test witness bunch.

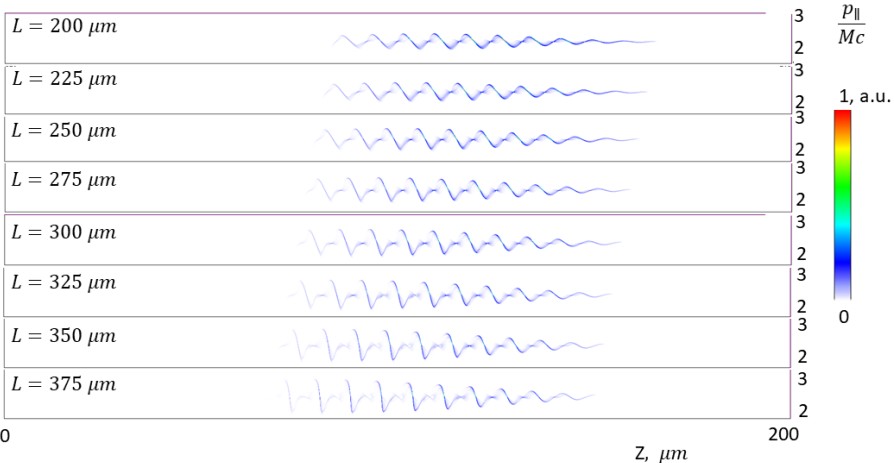

**Figure 4.** Phase space $\left( p_{\parallel}, z \right)$ of the witness bunch showing resonant acceleration/deceleration of the test particles. After the propagation distance of 375 μm, the most energetic muons start to overtake the decelerated particles. At this distance, tapering of the plasma channel would be required to further acceleration.

## 4. Discussion

In this paper, we show that the laser wakefield phase velocity can be efficiently controlled in a plasma channel. It is important that we use a beatwave of two laser pulses that are much longer than the plasma period. The frequency difference between these two laser pulses equals the local plasma frequency, so that the slow resonant excitation of the plasma wave is possible. Due to the fact that the driver energy is spread over many plasma periods, the interference pattern can run with an arbitrary velocity along the channel. This velocity is then controlled by the channel radius and the number of laser transverse modes excited in the channel. The laser channel mode with the higher transverse number has a higher longitudinal phase velocity than the mode with a lower transverse number. The wakefield velocity, however, is defined by the interference pattern of these two laser pulses, as given by Equation (8). By choosing the laser channel parameters, the plasma wake velocity can be matched exactly to the witness velocity. This can be the vacuum speed of light for ultra-relativistic witnesses, or slow subluminal velocities for low-energy weakly relativistic witnesses such as the muon bunches. In our example simulation, we matched the plasma wave velocity exactly to the initial muon bunch velocity, and show the resonant acceleration.

We describe here only the linear stage of acceleration. As the laser pulses interact with the wake over long distances, nonlinear processes will eventually modify them (change their frequency, local intensity, etc.) These nonlinear effects will naturally limit the useful channel length, thus, staging may be required. Furthermore, the velocity of a weakly relativistic witness will change as it is accelerated, leading to its dephasing. Solving this will require proper tapering of the channel parameters. These effects will be discussed elsewhere.

**Author Contributions:** Conceptualization, A.P.; analytical description, N.E.A., A.A.G., I.I.A., I.Y.K.; simulations A.P.; writing, original draft preparation, A.P.; writing, review and editing, A.A.G., I.Y.K. All authors have read and agreed to the published version of the manuscript.

**Funding:** This work has been supported by the Deutsche Forschungsgemeinschaft (Germany), by the Schwartz/Reisman Center for Intense Laser Physics (Israel), and by RFBR Projects No. 20-21-00150 and 20-52-12046 (Russia).

**Institutional Review Board Statement:** Not applicable.

**Informed Consent Statement:** Not applicable.

**Data Availability Statement:** Not applicable.

**Conflicts of Interest:** The authors declare no conflict of interest.

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
