# Peer review of "Laser–Plasma Wake Velocity Control by Multi-Mode Beatwave Excitation in a Channel"

_plasma, doi:10.3390/plasma6010003_

Round 1

Reviewer 1 Report

The paper proposed linear theory as well as PIC simulations for controlling the laser wake field phase velocity by multi-mode beatwave excitation in a channel is effective and feasible with good practicability.  However, there are some suggestions for this submission:

(1) The English language and style should be improved. 

(2) Some equations and the corresponding variables should be explained in detail.

(3) The section of "Conclusion" should be provided in this paper. 

Author Response

We thank the Referee for reading our manuscript carefully.
Here our answer to the comments:

Referee: (1) The English language and style should be improved. 

Reply: we have checked the manuscript and improved the English significantly.

Referee: (2) Some equations and the corresponding variables should be explained in detail.

Reply: We added missed explanation to the equations and variables.

Referee: (3) The section of "Conclusion" should be provided in this paper. 

Reply: The conclusions are given in the "Discussion" section.

Reviewer 2 Report

Comments to authors

A. Pukhov et al proposed that the laser wake field phase velocity can be efficiently controlled in a plasma channel. A beatwave of two long laser pulses is used. The frequency difference between these two laser pulses equals the local plasma frequency so that the slow resonant excitation of the plasma wave is possible. Because the driver energy is spread over many plasma periods, the interference pattern can run with an arbitrary velocity along the channel and generate the wake field with the same phase velocity. The proposed idea seems interesting but, in the article, the presented results and discussion should be explained more clearly.

Few comments

1)      What are the advantages of the proposed research?

2)      In the introduction they clearly explained the limitations of plasma-based accelerators, however in the results and discussion I could not see how they overcome them, simply presenting numerical values not much explanation.

3)      Could you please describe the shown data in Fig.2,3,4. To be honest I could not understand these patterns Discussion and abstract seem completely equal, better to write more clearly in the discussion part.

4)      Some typo:  Fig.1 color bar top value missed.

Author Response

e thank the Referee for commenting our manuscript.
Here our answer to the comments:

Referee: 1)  What are the advantages of the proposed research? 

Reply: the proposed method allows to adjust the plasma wake field phase velocity to the velocity of accelerated particles.
This is a significant advantage for plasma-based acceleration, because it lifts limitations set by dephasing between the plasma wave and the witness bunch.

Referee: 2)  In the introduction they clearly explained the limitations of plasma-based accelerators, however 
in the results and discussion I could not see how they overcome them, simply presenting 
numerical values not much explanation. 

Reply: In the results and discussion we show how we adjusted the wake field velocity to the subluminal velocity of the muon bunch. This is now clearly stated.

Referee: 3)  Could you please describe the shown data in Fig.2,3,4. To be honest I could not understand 
these patterns Discussion and abstract seem completely equal, better to write more clearly in 
the discussion part.   

Reply: We added captions to the figures describing the plotted data in more details and expanded "Discussion" section.

Referee: 4)  Some typo:  Fig.1 color bar top value missed.    

Reply: Fixed.

Round 2

Reviewer 2 Report

authors made the changes as per my comments; therefore I could suggest the manuscript should be accept in the present form.